# *Stelladens mysteriosus*: A Strange New Mosasaurid (Squamata) from the Maastrichtian (Late Cretaceous) of Morocco †

**Nicholas R. Longrich** [1,*] , **Nour-Eddine Jalil** [2,3] , **Xabier Pereda-Suberbiola** [4] **and Nathalie Bardet** [2]

1   Department of Life Sciences, University of Bath, Claverton Down, Bath BA2 7AY, UK
2   CR2P Centre de Recherche en Paléontologie—Paris, Muséum National d'Histoire Naturelle, CP38, 57 Rue Cuvier, 75005 Paris, France; nour-eddine.jalil@mnhn.fr (N.-E.J.); nathalie.bardet@mnhn.fr (N.B.)
3   Museum of Natural History of Marrakech, Faculty of Sciences Semlalia, Université Cadi Ayyad Marrakech, Marrakech 40000, Morocco
4   Departamento de Geología, Facultad de Ciencia y Tecnología, Universidad Del Pais Vasco/ Euskal Herriko Univertsitatea, Apartado 644, 48080 Bilbao, Spain; xabier.pereda@ehu.eus
*   Correspondence: nrl22@bath.ac.uk
†   (urn:lsid:zoobank.org:act:0ACF03C7-0A7A-4D92-9439-C89171AB70B1).

**Abstract:** Mosasaurids, a clade of specialized marine squamates, saw a major adaptive radiation in the Late Cretaceous, evolving a wide range of body sizes, shapes, and specialized tooth morphologies. The most diverse known mosasaurid faunas come from the late Maastrichtian phosphates of Morocco. Here, we report an unusual new mosasaurid, *Stelladens mysteriosus*, based on a partial jaw and associated tooth crowns from lower Couche III phosphatic deposits at Sidi Chennane, Oulad Abdoun Basin, Morocco. *Stelladens* is characterized by short, triangular tooth crowns with a series of strong, elaborate, and serrated ridges on the lingual surface of the tooth, functioning as accessory carinae. Morphology of the teeth and associated jaw fragment suggest affinities with Mosasaurinae. No close analogues to the unique tooth morphology of *Stelladens* are known, either extant or extinct. It may have had an unusual and highly specialized diet, a specialized prey-capture strategy, or both. The diversity of mosasaurid teeth is much higher than that of plesiosaurs, ichthyosaurs, or extant marine mammals, and likely reflects both the ecological diversity of mosasaurids and complex developmental mechanisms responsible for tooth formation in mosasaurines. Mosasaurid diversity continued to increase up to the Cretaceous–Paleogene boundary.

**Keywords:** Mosasauridae; marine reptiles; Cretaceous; Cretaceous–Paleogene mass extinction; Africa





## 1. Introduction

The Mosasauridae, a specialized family of marine lizards, underwent a major radiation in the Late Cretaceous to become the dominant marine predators of the day [1,2]. Their radiation followed an extended period of high turnover in the mid-Cretaceous, likely driven by the Cretaceous Thermal Maximum and a series of volcanic eruptions and associated oceanic anoxic events. These saw the extinction of large pliosaurids [3] and ichthyosaurs [4], followed by the appearance and diversification of the mosasaurids [1,5]. By the end of the Cretaceous, mosasaurids had undergone a remarkable radiation [1,6], as shown by their diverse jaw and tooth morphologies as well as the wide range of sizes [7], and had evolved to become apex predators [2].

The most diverse known mosasaurid assemblages are those of the Maastrichtian of the phosphates of Morocco. Here, trade winds pulled surface waters to the west, driving the upwelling of nutrient-rich bottom waters that fertilized the seas. High planktonic productivity supported a high biomass of small prey [8] which in turn fed a plethora of marine reptiles including plesiosaurs [9,10], marine turtles [11,12], rare crocodylians [13], and mosasaurids [7]. Mosasaurids are both diverse and abundant here, and are represented by four major clades, the Mosasaurinae [2,14–17], Halisaurinae [18,19], Tylosaurinae [20], and Plioplatecarpinae [21]. Other marine squamates found include Pachyvaranidae [22].

A striking feature of the mosasaurid fauna is the large number of rare species, many of small size. Whereas a handful of species, including *Thalassotitan atrox* [2], *Halisaurus arambourgi* [18], *Eremiasaurus heterodontus* [14], *Mosasaurus beaugei* [15], and *Gavialimimus almaghribensis* [21] account for the majority of the mosasaurids that have emerged, other taxa are uncommon, or occur very rarely. *Carinodens minalmamar* is known from a few isolated dentaries and teeth [17], and *Xenodens calminechari* is known from just one fragmentary maxilla [16]. Similar patterns are seen in modern marine mammal communities, in which a few species tend to be common, others are uncommon, and still others tend to be sighted only rarely [23–25], often as vagrants.

Recently, a striking new mosasaurid was discovered in the phosphates of Sidi Chennane, Oulad Abdoun Basin, Morocco (Figure 1). Known from only a partial dentary and two associated teeth, the specimen has a unique and astonishing tooth morphology not seen in any other mosasaurid, or other tetrapods. Here, a remarkable series of prominent, sharp and serrated ridges emerge from the lingual surface of the tooth and act as accessory carinae. The new mosasaurid suggests an unusual and specialized feeding strategy.

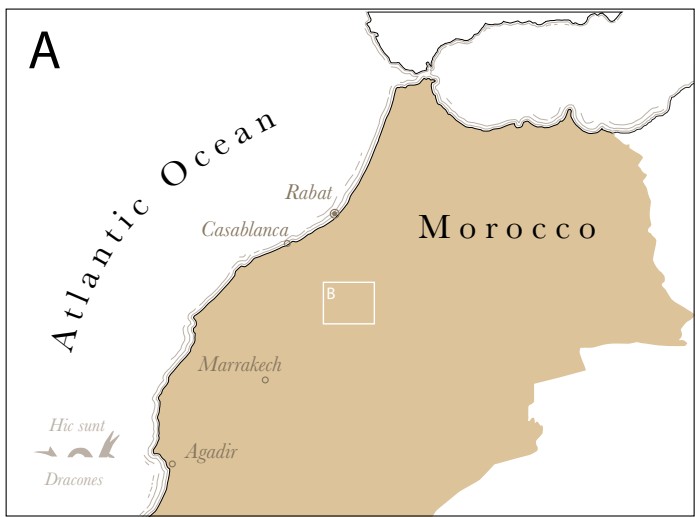

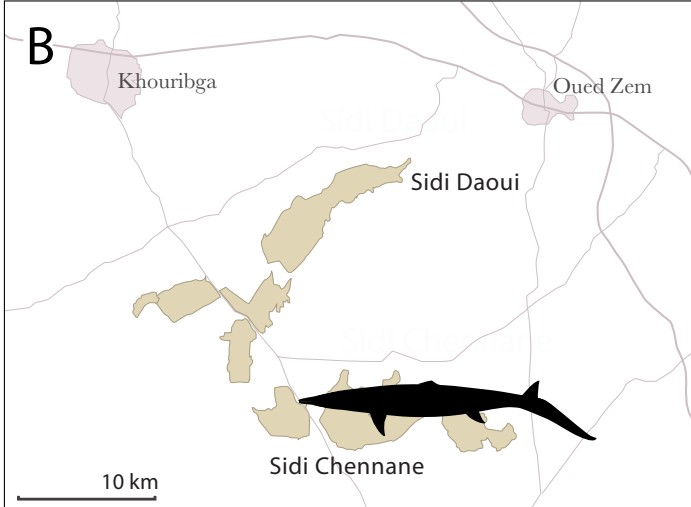

**Figure 1.** Map of the central part of Morocco showing (**A**), the location of the phosphate mines of the Oulad Abdoun Basin, and (**B**), the type locality of *Stelladens mysteriosus* in Sidi Chennane, south Oulad Abdoun Basin.

## 2. Geological Setting

The phosphates of Morocco are part of a large belt of sedimentary rocks extending from the Middle East to North and West Africa, and up to the Pernambuco province of

Brazil [26]. They were laid down in a shallow seaway, when a period of high sea levels saw the eastern Atlantic Ocean flood much of north Africa.

The phosphates of Morocco span the Late Cretaceous and Early Paleogene, and are the most stratigraphically expanded of all these phosphatic deposits [26]. They are economically exploited in two main basins, the Oulad Abdoun and the Ganntour basins (Figure 1).

In the Oulad Abdoun Basin, from which most of the vertebrate specimens have been unearthed, the phosphates are divided into several beds or Couches [27]. From bottom to top these are Couche III, which is Maastrichtian in age, and Couche II, I and 0, which are Paleogene (Danian to Ypresian) (Figure 2).

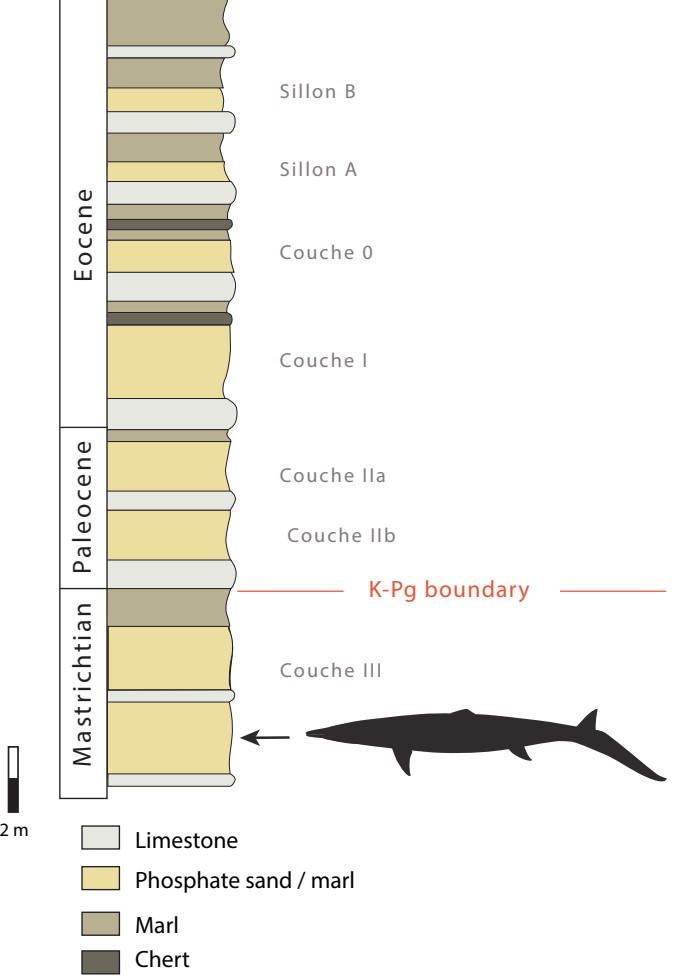

**Figure 2.** Synthetic stratigraphic column of the Late Cretaceous—Paleogene phosphatic series in the Oulad Abddoun Basin of Morocco, showing position of *Stelladon mysteriosus* in lower Couche III.

Couche III consists of phosphatic sands, marls, and limestone, with several highly fossiliferous bonebeds. The upper part of Couche III is latest Maastrichtian in age on the basis of shark fossils and chemostratigraphy [27,28]. The lower part of Couche III contains a broadly similar fauna but with distinct species, suggesting it is only slightly older.

The vertebrate remains found here are extremely abundant and represented mainly by marine taxa [29], including dozens of selachian species [30], osteichthyans [31], as well as more than fifty marine reptile species, including plesiosaurs, sea turtles, and mosasaurid squamates [29]. Pterosaurs are also present [32,33], as well as rare dinosaurs [34–36]. Isolated teeth and bones are extremely abundant, but rarer associated and articulated remains, including some mostly complete skeletons, are also present.

Remains often show disarticulation [2], resulting from the decomposition of floating carcasses, scavenging, and/or predation. Some specimens are articulated in near-life position with pristine preservation of the bone, suggesting rapid burial and limited scavenging. Others, however, show extensive disarticulation and bone surfaces that have been heavily modified, suggesting scavenging by a benthic community, similar to modern whalefalls.

## 3. Results

### 3.1. Systematic Paleontology

Squamata Oppel, 1811 [37].
Mosasauridae Gervais, 1852 [38].
Mosasaurinae Gervais, 1852 [38].
*Stelladens mysteriosus* new genus and species.

*Etymology.* The genus' name is from the Latin *stella*, 'star' + *dens*, tooth. The species' name is from the Latin *mysterium*, 'mystery', because of the mysterious structure of the teeth.

*Holotype.* MHNM.KHG.1436, partial left dentary and two associated teeth (Figures 3–5).

*Locality and Horizon.* The holotype comes from Sidi Chennane phosphate mine, in the Oulad Abdoun Basin, Khouribga Province, Morocco. It was recovered from the lower part of Couche III (Figures 1 and 2). Upper Couche III is dated to the late Maastrichtian on the basis of shark teeth [30], and to the latest Maastrichtian based on carbon and oxygen chemostratigraphy [27]. The age of lower Couche III is not well-constrained. Lower Couche III appears to contain a broadly similar fauna, but taxa are distinct at the species level, suggesting it is somewhat older than Upper Couche III.

*Diagnosis.* Mosasaurine mosasaurid characterized by the following unique character combination: low, triangular, weakly recurved crowns with a strong U-shaped cross-section; two prominent serrated carinae, the posterior one being more marked and pinched"from the main shaft; labial surface almost flat bearing 6–8 subtle low ridges; lingual surface strongly convex and bearing 2 to 4 very prominent, sharp and serrated ridges.

### 3.2. Comparative Description

#### 3.2.1. General Remarks

The specimen shows extensive surface damage to the bone, suggesting a period of exposure on the seabed prior to burial. It is covered by a calcareous concretion.

#### 3.2.2. Dentary

The dentary (Figure 3) is broken anteriorly and posteriorly. Although the bone is obscured by damage and adhering matrix, it can be identified as a dentary on the basis of a convex lateral surface (the maxilla tends to have a flatter lateral surface, often with a ridge above the teeth), as well as grooves on the medial surface, representing the top of the Meckelian canal and the splenial articulation surface. In lateral view, the ventral edge of the bone is straight. The medial parapet extends as high as the lateral one, a derived condition found in Mosasaurinae [39]. On the lateral surface, below and between the dental alveoli, occlusal pits for maxillary teeth are visible, as in certain mosasaurines, such as *Mosasaurus hoffmanni* [40] and *Eremiasaurus heterodontus* [14]. The jaw is gently bowed inward in dorsal view; the wider end can be assumed to represent the posterior end of the bone, making this a left dentary.

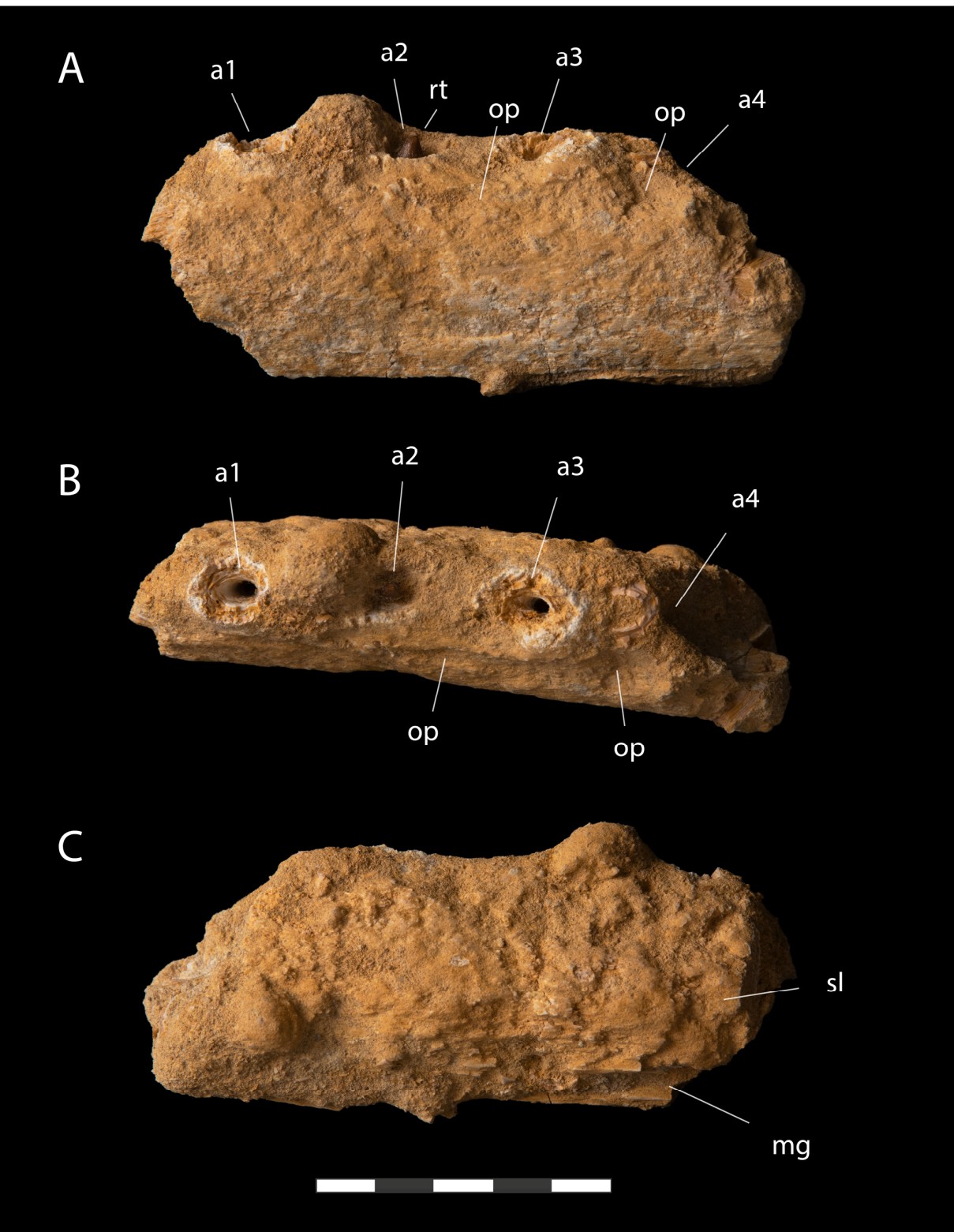

**Figure 3.** *Stelladens mysteriosus*, MHNM.KHG.1436, holotype, partial left dentary, Sidi Chennane, Oulad Abdoun Basin, Morocco, Lower Couche III, Maastrichtian. In (**A**), lateral view; (**B**), dorsal view; (**C**), medial view. Abbreviations: a1-a4, first through fourth preserved alveoli; mg, Meckelian groove; op, occlusal pits for maxillary teeth; sl, subdental lamina; sp, splenial facet. Scale = 5 cm.

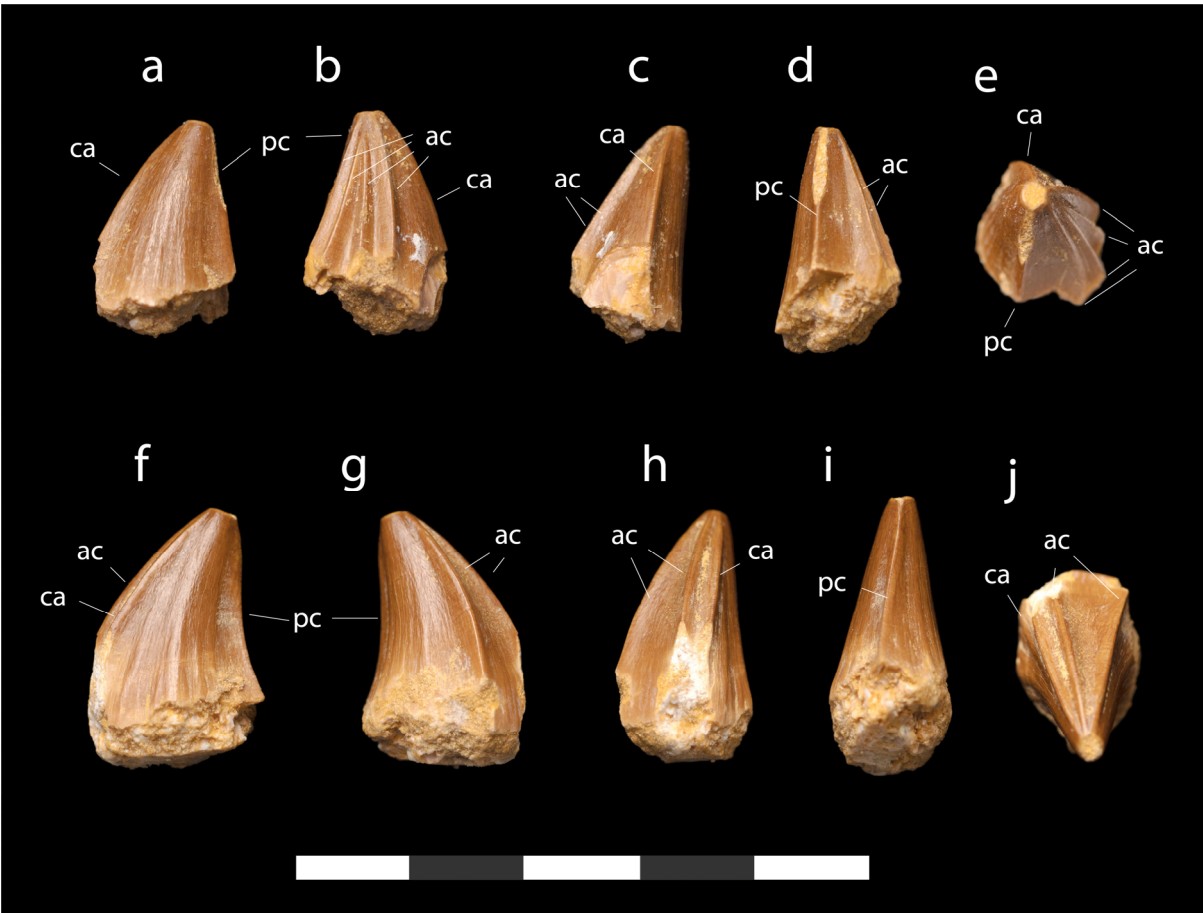

**Figure 4.** *Stelladens mysteriosus*, MHNM.KHG.1436, holotype, teeth, Sidi Chennane, Oulad Abdoun Basin, Morocco, Lower Couche III, Maastrichtian. In (**a**,**f**), labial view, (**b**,**g**), lingual view, (**c**,**h**), anterior view, (**d**,**i**), posterior view, (**e**,**j**), occlusal view. Abbreviations: ac, accessory carinae; ca, anterior carina, pc, posterior carina. Scale = 5 cm.

Four tooth positions are preserved. The anteriormost tooth is broken at the base of a circular crown; the second tooth position includes the apex of an emerging tooth; the third one is broken at the base of a more elliptical crown; the last one is hollow. The breaks of the second and fourth teeth are fresh, indicating that they were broken during excavation. The basal sections of the broken teeth confirm the dentary fragment as belonging to the left mandibular ramus.

Comparisons with *Mosasaurus* suggest the dentary was perhaps 40–50 cm long, and the skull around 80 cm long. Assuming proportions similar to *Eremiasaurus*, *Stelladens* may have grown to around 5 m in length (Figure 6).

### 3.2.3. Teeth

Two basally broken tooth crowns were recovered in association with the dentary (Figures 4 and 5). They are similar in diameter to the broken teeth of the dentary fragment, but do not cleanly attach to it, perhaps due to damage.

The smaller crown has a more subcircular section than the larger tooth, which has a more elliptical section. This suggests that the smaller tooth was located more anteriorly in the jaw. However, they both exhibit roughly anteroposteriorly aligned anterior and posterior main carinae, indicating that they both come from near the middle of the jaw.

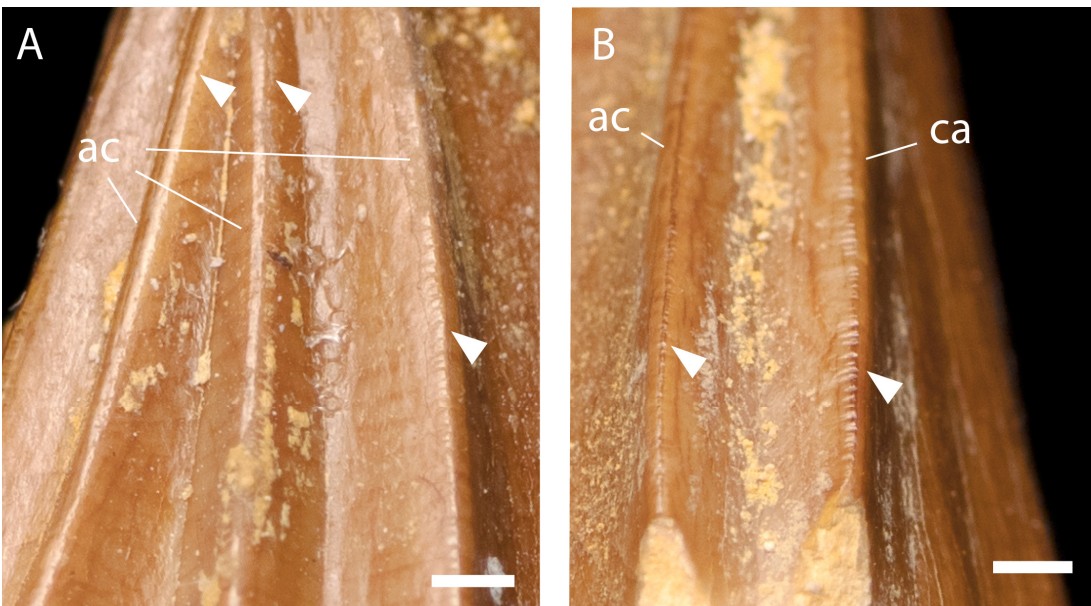

**Figure 5.** Close-up showing the accessory carinae of the first (**A**) and second (**B**) tooth, and serrations (arrows). Abbreviations: ac, accessory carinae; ca, anterior carina.

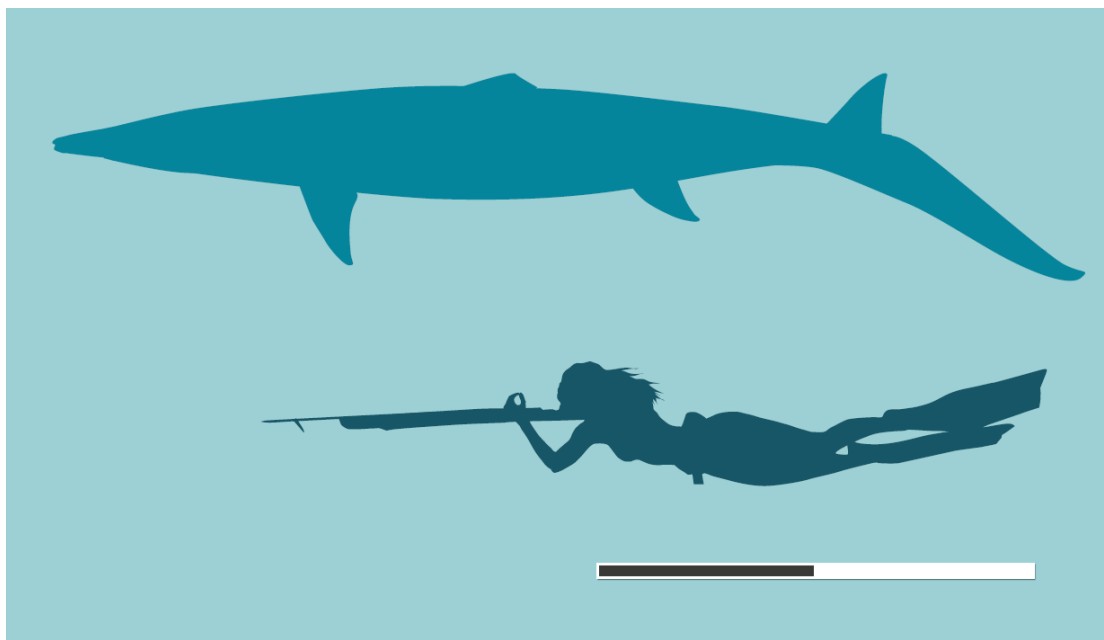

**Figure 6.** Silhouette showing the approximate size of *Stelladens mysteriosus*. Scale = 2 m.

The smaller and more anterior tooth crown (Figure 4a–e) has a short, almost straight, triangular shape in lateral view. It has a slightly convex anterior edge and a straighter posterior one; it is roughly equal in anteroposterior and labio-lingual width. Anterior and posterior carinae, aligned anteroposteriorly, are present. Both are prominent, especially the distal carina which appears as pinched from the main shaft of the crown, and serrated, as is commonly seen in Mosasaurinae [41]. The labial surface of the tooth is almost flat, and the lingual surface is strongly convex, giving the crown a U-shaped section as in *Mosasaurus* [41,42]; it has a roughly triangular-to-circular basal cross-section. The labial surface of the tooth is subtly ridged; it bears 6–8 low ridges separated by shallow grooves. Basal striations, of the sort seen in Tylosaurinae and Plioplatecarpinae [42], are absent.

The lingual surface of the tooth bears a series of four very prominent ridges, not observed before in any mosasaurid. These ridges have a triangular section, with a sharp edge, and are separated by deep, V-shaped grooves. The ridges bear serrations as in the carinae, which appears to be unique among Mosasauridae. The anteriormost ridge has serrations that are well-developed and extend far down the ridge. The more posterior ridges have more subtle serrations that are developed apically, while the ridges are unserrated basally (Figure 5A).

The apex of the tooth bears a flat, ovoid wear facet, which suggests the tooth repeatedly impacted against hard surfaces. The posterior carina bears a prominent, rough wear facet located on its upper third, suggesting it also ground against hard prey items. The surface of the tooth is ornamented by subtle, anastomosing ridges as in prognathodontine mosasaurines [42], giving it a silky aspect.

The second, larger, and more posterior tooth crown (Figure 4f–j) is again short, triangular, and weakly recurved in lateral view. It has a convex anterior edge and a concave posterior one, giving the tooth a more hooked shape than the first tooth. It is slightly larger anteroposteriorly than labiolingually. As in the first tooth, there are two strong carinae, roughly aligned anteroposteriorly. Again, as in the first tooth, there is a relatively flat labial surface and a strongly convex lingual surface. The labial surface bears five or six ridges, several of which are prominent and extend far up the crown, while the others are subtle and disappear relatively low on the tooth; they are separated by shallow gutters and grooves.

As in the first tooth crown, the lingual surface bears prominent, sharp ridges on the anteromedial surface, but instead bears only two. These ridges are tall and blade-like, and are separated by deep grooves and gutters. The first ridge or accessory carina bears well-developed serrations along its length (Figure 5B); the second ridge has serrations apically, but not basally. Four or five more subtle ridges are developed posteromedially.

As in the first tooth crown, the surface of the tooth is ornamented with anastomosing ridges, giving it a silky aspect. The occlusal wear facet is planar, and there is faint wear of the posterior carina near the apex of the tooth.

## 4. Discussion

### 4.1. Comparison and Identification

The holotype specimen of *Stelladens* is highly fragmentary, and diagnosed on the basis of tooth morphology only. Generally speaking, mosasaurid teeth may be diagnostic to the genus level, and sometimes to the species level. The remarkable morphology seen here—unique among squamates, or even tetrapods—distinguishes *Stelladens* from all other known mosasaurids or squamates, however, validating its recognition as a distinct genus and species.

The limited nature of the material hinders comparisons, but the material suggests affinities with Mosasaurinae. Most other mosasaurids, including Halisaurinae, Plioplatecarpinae, and Tylosaurinae, have relatively simple teeth [42]. Those of halisaurines are small, recurved, or hooked cones, with an ornamentation consisting of very fine, anastomosing ridges, and lack serrations [18,19,42,43]. Plioplatecarpine teeth are spike-like or hook-like, with numerous ridges and basal striae, and also lack serrations [21,42]. Tylosaurine teeth are spike-like with ridges and striations [42] or, in the case of *Hainosaurus*, have blade-like sections and strong carinae, with fine serrations [20]. None of these tooth morphologies approach the condition seen here.

Several features of the teeth suggest affinities with Mosasaurinae, which also exhibit greater plasticity and a much larger range of morphological diversity, reflecting their higher diversity compared to other subfamilies, especially during the Maastrichtian [7]. In *Stelladens*, the teeth are more or less U-shaped, with a relatively flat labial surface and a strongly convex lingual surface, similar to *Mosasaurus* [40,41]. The prominent carinae are also typical of Mosasaurinae, as are the serrations, which are seen for example in *Mosasaurus hoffmanni* (NRL, pers. obs.), *Eremiasaurus* [14], *Thalassotitan* [2], and some *Prognathodon* species (NB, pers. obs.), etc. The accessory ridges and grooves separating them

may be an elaboration of the prisms seen in Mosasaurini. The rugose sculpturing of the enamel is similar to that seen in certain mosasaurines [42], particularly *Thalassotitan* [2] and other Prognathodontini; however the enamel of other mosasaurines, such as *Mosasaurus*, is smoother.

### 4.2. The Diet of Stelladens

The specialized dentition of *Stelladens* implies a specialized diet, a specialized mode of prey capture, or perhaps both. The lack of close analogues among modern reptiles or any extant tetrapod makes it difficult to infer function, however.

Many aquatic or piscivorous tetrapods, including many mosasaurid squamates, ichthyosaurs, plesiosaurs, crocodylians, spinosaurid dinosaurs, and pterosaurs, have prominent ridges on the teeth [44,45], which probably function to help in catching and handling fish and other prey [44]. *Mosasaurus*, for example, has ridges on the teeth, and fish found in the gut confirm their piscivorous habits [46]. Similar arguments hold for animals preying on soft-bodied cephalopods; ichthyosaurs with cephalopods as stomach contents have ridged teeth [45]. The function of these ridges is not well-understood; presumably, they help the tooth pierce the prey item or help hold it fast in the jaws.

Yet, no known fish-eater has anything approaching the elaborate accessory carinae seen in *Stelladens*; if this design functions for catching fish or squid, it seems curious that nothing else has solved the problem in the same way. Other aspects of the tooth's structure seem poorly adapted for catching fish or soft-bodied cephalopods. The teeth are also relatively short and stout, which would have given them considerable strength in bending, whereas most fish eaters or squid eaters have relatively tall and slender crowns [45], acting to pierce and hold small prey. The anastomosing texture of the enamel is also unusual. Assuming *Stelladens* was a piscivore or squid-eater, then it seems to have hunted unusual fish or unusual cephalopods, or had an unusual way of catching them.

It seems unlikely the teeth were used to process thick-shelled prey such as bivalves or echinoids. In the durophagous mosasaurids *Globidens* [47] and *Carinodens* [17,48], tooth crowns have short, blunt apices to prevent fracturing of the tips. The enamel in those species is also thick and highly rugose to reduce chipping or spalling, and the teeth often exhibit heavy wear. None of these features occur in *Stelladens*. However, it is conceivable that the ridges on the teeth might have functioned to fracture thinner-shelled species such as crustaceans, for example, shrimp and lobsters, or thin-shelled ammonites. Furthermore, the apical wear seen in the teeth suggests that prey were not exclusively soft-bodied.

### 4.3. Implications for Mosasaurid Evolution

*Stelladens* adds to the remarkable morphological diversity and plasticity of mosasaurid teeth [7]. These include conical spikes and hooks to spear and hold small prey [19]; larger, spike-like teeth to impale large prey [40]; blade-like teeth to cut larger prey items [16]; conical teeth to tear very large prey [2]; and a diversity of blunt and molariform teeth adapted to crush hard-shelled prey such as mollusks, echinoids, and crustaceans [17,47,49]. Mosasaurid teeth also show adaptations to withstand high stress in the tooth microstructure and possibly tooth chemistry [50].

The diversity of mosasaurid teeth stands in marked contrast to the more limited morphological diversity seen in modern analogues, extant cetaceans and pinnipeds. Although mammal teeth reach a level of complexity not seen in mosasaurids, mosasaurid teeth arguably exhibit a wider range of morphologies and perform a wider range of functions than the teeth of marine mammals. Mosasaurid tooth diversity also includes many morphologies not seen in modern squamates. This high morphological, ecological, and functional diversity was part of a major mosasaur radiation that preceded the end-Cretaceous mass extinction, suggesting that mosasaurid diversity remained high or even increased just prior to their extinction at the K-Pg boundary [50]. This marine radiation was in turn part of a much broader radiation of Cretaceous squamates, which saw high diversity in teeth and jaws [51–53] in the Late Cretaceous.

How the unusual teeth of *Stelladens* evolved is unclear. Split or duplicated carinae appear as development abnormalities in dinosaurs [54] and other archosaurs [55], and in sabertooth cats [56]. However, the fact that two teeth both exhibit these accessory carinae, and that they are developed in the same way—with serrations better developed anteriorly than posteriorly— suggests that it is not a pathology. Additionally, if *Stelladens* was a pathological individual rather than a distinct species, one would expect the teeth to resemble a known taxon. However, the tooth shape does not closely resemble any known species. It seems most parsimonious to assume *Stelladens* is a distinct taxon and that this is the natural morphology of this species.

The accessory ridges in *Stelladens* resemble carinae in being elaborated to form a prominent cutting edge, and in bearing serrations indistinguishable from those of anterior and posterior carinae. The close resemblance of ridges to the carinae suggests that they may be homologous to carinae. However, these accessory ridges occupy the same position as the ridges of the lingual facets or 'prisms' [42] in related Mosasaurinae. Their position therefore suggests that they could be homologous with prisms. It is also conceivable that the genes responsible for the formation of carinae also control the formation of facets, in which case the lingual ridges of *Stelladens* might at some level be homologous with both carinae, and with facets.

Whether anything similar exists in any other tetrapod remains unclear. The lizard *Peneteius* evolved accessory tooth cusps [57] (and the carinae of mosasaurids likely evolved from and are homologous with the cusps of other lizards). This implies that the duplication of tooth structures is possible in lizard teeth, but the arrangement and presumed function in the two would have been very different. Plesiosaur teeth, including Leptocleididae [45,58] and Pliosauridae [45,59,60], bear enamel ridges that resemble those forming the cutting edge of the carinae of the teeth, and these furthermore may have small knots or denticle-like structures. Herbivorous dinosaurs such as Hadrosauridae and Ceratopsidae evolved prominent lingual ridges of the teeth [61,62], but these lack denticles, and it is unclear whether they are homologous to the tooth carinae, or a distinct structure.

### 5. Conclusions

A partial mosasaurid jaw and the associated teeth exhibit a highly unusual morphology, with two to four prominent, sharp, and serrated ridges on the lingual surface of the teeth. This unusual morphology diagnoses a new genus and species, *Stelladens mysteriosus*, and underscores the high morphological and ecological diversity of mosasaurids in the latest Cretaceous, especially among Mosasaurinae [7]. *Stelladens* presumably had a specialized feeding strategy, but the lack of any extinct or modern analogues makes it difficult to infer its ecology. As part of their radiation, mosasaurids experimented with unusual tooth morphologies, similar to dinosaurs and mammals. Mosasaurid diversity was high in the Maastrichtian and continued to increase prior to their extinction at the K-Pg boundary.

**Author Contributions:** Conceptualization, N.R.L.; investigation, N.R.L., N.B., N.-E.J. and X.P.-S.; writing—original draft preparation, N.R.L. and N.B.; writing—review and editing, X.P.-S. and N.-E.J. All authors have read and agreed to the published version of the manuscript.

**Funding:** Research of XPS is financed by the Spanish Ministry of Science and Innovation (MCIN) and the European Regional Development Fund (FEDER) (research project PID2021-122612OB-I00), and by the Basque Country Gouvernement (research group IT1485-22). This study is carried out within the framework of the agreement between the universities of Bath and Cadi Ayyad.

**Data Availability Statement:** All data used in the study are included in the paper.

**Acknowledgments:** Thanks to Mustapha Meharich for assistance to NL in Morocco and to the reviewers for their comments.

**Conflicts of Interest:** The authors declare no conflict of interest.

## Abbreviations

MHNM: Muséum d'Histoire Naturelle de Marrakech, Université Cadi Ayyad, Morocco.

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
