# Peer review of "Stelladens mysteriosus: A Strange New Mosasaurid (Squamata) from the Maastrichtian (Late Cretaceous) of Moroccoâ€"

_2813-6284, doi:10.3390/fossils1010002_

Round 1
Reviewer 1 Report
The article describes a new and unusual fragmentary dentary and closely associated teeth from a mosasaur from Morocco. The new name offered for the species is descriptive and appropriate for the material presented. I enjoyed reading the article which was well written and a useful addition to the growing mosasaur fauna of the upper Maastrichtian from the phosphate deposits of the Oulad Abdoun Basin.
There were only two potential areas that may benefit further research but understand that the lack of material would not allow for destructive analysis of the teeth. Perhaps this is something that would benefit from further discoveries in the future. I felt that it might be possible to identify the teeth as definitely belonging to the mosasaurs based on their histology. Street et al. 2021 indicated that the histology of striations is unique amongst the mosasaurs and may be a useful means of differentiating them at the ordinal or subordinal level.
The other area of potential future research is relating to the taphonomy that is alluded to in the article where fossil microborings in the dentine may offer further information on the post-mortem biodegradation and scavenging by organisms on the sea floor (see Owocki & Madzia 2020). This may provide stronger comparisons with modern whalefalls indicated on line 105.
I am uncertain whether the serrations on the carinae can be used as diagnostic characters as these are not the same as serrations in other tetrapods. In mosasaurs, the ‘serrations’ appear to be caused by variations in the thickness of the enamel and lack a dentine core. This would mean that the ‘serrations’ would wear down to leave no evidence of their prior existence in mosasaurs, so their absence would not necessarily mean a lack of ‘serrations’, or ‘false denticulations’ in the younger tooth (Street et al. 2021).
There were a few errors that will need to be addressed and suggestions, but nothing major.
Line 34. Delete ‘The’
Line 37. “Thermal Maximum, a series of volcanic…” (delete ‘and’)
Line 82. “Maastrichtian in age, and Couche II, I and 0…” (add ‘and’)
Line 87. “it is only slightly older.” (add ‘only’ – wouldn’t the lower part be older by superposition?)
Line 94. “…are also present.” (change ‘often’ for ‘also’ – I felt that something can’t be rarer and often present)
Line 116-117. I am not a Latin scholar sadly so not sure about the mixing of feminine, neuter, and masculine Latin in binomials. The name looks good though.
Line 131. Note above relating to serrations.
Line 441-442. Remove one of the “Proceedings of the” and italicize the journal name – if necessary?
Owocki & Madzia 2020. Predatory behaviour in mosasaurid squamates inferred from tooth microstructure and mineralogy. Cretaceous Research, 111. https://www.sciencedirect.com/science/article/pii/S0195667119303106
Street et al. 2021. A histological investigation of dental crown characters used in mosasaur phylogenetic analyses. Vert. Anat. Morph. Palaeo. 9. https://journals.library.ualberta.ca/vamp/index.php/VAMP/article/view/29372
Author Response
Response to reviewer 1
The article describes a new and unusual fragmentary dentary and closely associated teeth from a mosasaur from Morocco. The new name offered for the species is descriptive and appropriate for the material presented. I enjoyed reading the article which was well written and a useful addition to the growing mosasaur fauna of the upper Maastrichtian from the phosphate deposits of the Oulad Abdoun Basin.
There were only two potential areas that may benefit further research but understand that the lack of material would not allow for destructive analysis of the teeth. Perhaps this is something that would benefit from further discoveries in the future. I felt that it might be possible to identify the teeth as definitely belonging to the mosasaurs based on their histology. Street et al. 2021 indicated that the histology of striations is unique amongst the mosasaurs and may be a useful means of differentiating them at the ordinal or subordinal level.
We’ve added references to Street and note that absence of basal striae argue against affinities with Plioplatecarpini or Tylosaurinae. Referral to Mosasaurinae seems fairly well-supported based on multiple characters:
Strong carinae
Serrations of carinae
U-shaped tooth cross section
Occlusal pits for maxillary teeth on dentary
Anastomosing ornament
We have at least five characters which all suggest mosasaurine affinities, so we feel fairly confident in this referral. Histology is interesting in its own right and more work on mosasaur histology would be of interest to understand tooth evolution but beyond the scope of the paper— the specimens are currently in Morocco and we don’t currently have a setup to do histology.
The other area of potential future research is relating to the taphonomy that is alluded to in the article where fossil microborings in the dentine may offer further information on the post-mortem biodegradation and scavenging by organisms on the sea floor (see Owocki & Madzia 2020). This may provide stronger comparisons with modern whalefalls indicated on line 105.
Again, this would be interesting in its own right. There are clearly a lot of complicated taphonomic processes operating in the phosphates- skeletons are disarticulated by scavengers, show toothmarks, some are digested by mosasaurs, bones show borings. However this is unfortunately beyond the scope of the current paper, which is focused on taxonomy and not taphonomy.
I am uncertain whether the serrations on the carinae can be used as diagnostic characters as these are not the same as serrations in other tetrapods. In mosasaurs, the ‘serrations’ appear to be caused by variations in the thickness of the enamel and lack a dentine core. This would mean that the ‘serrations’ would wear down to leave no evidence of their prior existence in mosasaurs, so their absence would not necessarily mean a lack of ‘serrations’, or ‘false denticulations’ in the younger tooth (Street et al. 2021).
Its possible to study erupting teeth to find teeth without wear. So far no one has observed these denticles on teeth of any other mosasaur, although its possible people haven’t looked closely enough.
There were a few errors that will need to be addressed and suggestions, but nothing major.
Line 34. Delete ‘The’
Done
Line 37. “Thermal Maximum, a series of volcanic…” (delete ‘and’)
The Cretaceous Thermal maximum and the eruptions/anoxia are distinct-but-related events, we’d argue the thermal maximum and anoxia presumably being triggered by eruption, although other events come into play in thermal maxima, e.g. methane release, orbital forcing, etc.
Line 82. “Maastrichtian in age, and Couche II, I and 0…” (add ‘and’)
Added
Line 87. “it is only slightly older.” (add ‘only’ – wouldn’t the lower part be older by superposition?)
Added
Line 94. “…are also present.” (change ‘often’ for ‘also’ – I felt that something can’t be rarer and often present)
added
Line 116-117. I am not a Latin scholar sadly so not sure about the mixing of feminine, neuter, and masculine Latin in binomials. The name looks good though.
Technically, there are guidelines but no rules in taxonomy, any combination of the 26 characters in any language, or no language, e.g. ‘Mxyzptlk’ would work...
Line 131. Note above relating to serrations.
We’d argue that the serrations might suffer wear, but in unerupted or newly erupted teeth they can be assessed, similar to how hadrosaurs like Ajnabia have diagnostic characters of the teeth, which can only be assessed by looking at unerupted teeth.
Line 441-442. Remove one of the “Proceedings of the” and italicize the journal name – if necessary?
Fixed
Owocki & Madzia 2020. Predatory behaviour in mosasaurid squamates inferred from tooth microstructure and mineralogy. Cretaceous Research, 111. https://www.sciencedirect.com/science/article/pii/S0195667119303106
Street et al. 2021. A histological investigation of dental crown characters used in mosasaur phylogenetic analyses. Vert. Anat. Morph. Palaeo. 9. https://journals.library.ualberta.ca/vamp/index.php/VAMP/article/view/29372
Both references added
Reviewer 2 Report
This contribution is dedicated to the description of a new mosasaurid from the Maastrichtian of Morocco. Despite its fragmentary nature the holotype and only specimen is indeed of great interest, demonstrating a peculiar morphology of tooth crowns. It is an important contribution to our knowledge of the diversity and disparity of mosasaurids. The manuscript, however, appears somewhat underelaborated.
General comments
The statement that the new taxon has several additional carinae with serrated edges is phenomenal and indeed implies the first record of a vertebrate animal with that many serrated ridges reaching the apex and not branching. If true, this discovery is an important contribution to our knowledge of the phenomenon of accessory carinae. However, I recall the “Sagan standard” here: “Extraordinary claims require extraordinary evidence”. Well, in the case of the present paper, simple and ordinary evidence is missing. The authors provide only general views of the teeth, so it is not possible to observe any serrations of the main carinae, not to mention the additional ones. I recommend adding magnified views of the main and accessory carinae to demonstrate the serration morphology. Otherwise, it seems like the authors expect the reader to take their claims on faith, which is not a good way in modern science.
Furthermore, no explanation of the formation of this peculiar morphology is proposed by the authors, and no other vertebrate with reminiscent morphologies is discussed. Particularly I recommend the authors consider the existing literature on supernumerary carinae in other vertebrates. Useful discussions are provided by e.g. Beatty & Heckert (2009 https://www.tandfonline.com/doi/abs/10.1080/08912960903154511 ) and by Welsh et al. (2020 - https://www.sdaos.org/wp-content/uploads/pdfs/2020/20-14%20Welsh%20full.pdf).
Another question, which remains unanswered – how are these carinae related to the edges between facets on teeth, known in other mosasaurid taxa? Superficially similar teeth are seen in the Moroccan Mosasaurus beaugei, just make the edges between the facets more protruding and facet surfaces more concave… It is important to demonstrate that these ridges are indeed additional serrated carinae, but not flutes, like those of some theropod dinosaurs (Paronychodon etc.)
It is unclear why the second tooth is obliquely photographed in apical view so that its cross-section and pattern of ridges and carinae distribution are incompletely seen.
Furthermore, in the second tooth, there are only two accessory carinae (vs four in the first) – this variation should be discussed.
In summary, I recommend:
- adding close-up figures of serrations on both main and accessory carinae.
- adding remarks on the variation of the number of accessory carinae and their distribution around crown circumference
optional - adding discussion on supernumerary carinae in the broader context of other vertebrates
Specific comments
Abstract. Line 27: “diversity of mosasaurid teeth is much higher than that of plesiosaurs, ichthyosaurs, or marine mammals…” – I cannot agree regarding marine mammals - basal cetaceans have teeth of complex morphology with multiple accessory denticles and complex enamel ornamentation; furthermore the teeth of some pinnipeds, like crabeater seal, have a very complex morphology, clearly unparalleled by mosasaurs.
Line 34: extra The in the first sentence
Line 61: Some reference is needed here to support the authors' statement e.g. https://doi.org/10.1073/pnas.110152510 or anything else appropriate to the context.
Line 188: add reference where the “silky” enamel surface of prognathodontine mosasaurines is described.
Lines 195-199: Correct figure captions, commas, tot dashes between the labels otherwise you have labial views on all pictures between a–f. and throughout. Also anterior, not “anteiro”.
Figure 5 is uninformative, as it is pure speculation based on a very fragmentary specimen. I recommend not including it in the MS
Line 219. Format of the section heading.
Line 255: correct to numerical citation. Also, the cited paper is not relevant to the context, as Zverkov & Pervushov (2020) described only a vertebra, and wrote nothing about teeth. Probably, the authors wanted to cite another paper, on pliosaurid teeth - https://doi.org/10.1111/pala.12367
Line 271-272: “such as” repeated. Rephrasing recommended
Line 281: “diversity of mosasaurid teeth stands in marked contrast to the more limited morphological diversity seen in their modern analogues, cetaceans and pinnipeds…” – this is not entirely true, as also commented on the abstract above - basal cetaceans have teeth of complex morphology with multiple accessory denticles and complex enamel ornamentation; furthermore the teeth of some pinnipeds, like crabeater seal, have a very complex morphology, clearly unparalleled by mosasaurs.
So authors can say that mosasaurids occupied more feeding guilds than modern marine mammals, but they should also acknowledge that there are tooth morphotypes and feeding adaptations among modern mammals not paralleled by mosasaurs.
Line 287: Title caption. I suppose ‘Conclusions’ should be here.
Author Response
The statement that the new taxon has several additional carinae with serrated edges is phenomenal and indeed implies the first record of a vertebrate animal with that many serrated ridges reaching the apex and not branching. If true, this discovery is an important contribution to our knowledge of the phenomenon of accessory carinae. However, I recall the “Sagan standard” here: “Extraordinary claims require extraordinary evidence”. Well, in the case of the present paper, simple and ordinary evidence is missing. The authors provide only general views of the teeth, so it is not possible to observe any serrations of the main carinae, not to mention the additional ones. I recommend adding magnified views of the main and accessory carinae to demonstrate the serration morphology. Otherwise, it seems like the authors expect the reader to take their claims on faith, which is not a good way in modern science.
The serrations on the carinae are in fact visible in Figure 4. One simply has to zoom in on the image. However, we have provided a second figure which provides a zoomed-in view and labels the serrations.
Furthermore, no explanation of the formation of this peculiar morphology is proposed by the authors,
Understanding the formation of these structures would require understanding how patterns of gene expression are evolving. Presumably:
(1a) genes that control the formation of the carina are expressed on the lingual surface of the tooth (ridges are homologous with carinae), or,
(1b) the expression of genes that control the formation of the facets/prisms are expressed to elaborate these structures (ridges are homologous with prisms/facets)
And also
(2) genes that cause serrations to form on the carinae are expressed along the ridges.
However trying to understand the evo-devo of an extinct mosasaur is speculative and a bit beyond the scope of what we’re trying to do.
and no other vertebrate with reminiscent morphologies is discussed. Particularly I recommend the authors consider the existing literature on supernumerary carinae in other vertebrates. Useful discussions are provided by e.g. Beatty & Heckert (2009 https://www.tandfonline.com/doi/abs/10.1080/08912960903154511 ) and by Welsh et al. (2020 - https://www.sdaos.org/wp-content/uploads/pdfs/2020/20-14%20Welsh%20full.pdf).
these are pathological- which our animal does not seem to be- but we have added these and other references to split carinae in discussing why we think the animal isn’t pathological
Another question, which remains unanswered – how are these carinae related to the edges between facets on teeth, known in other mosasaurid taxa? Superficially similar teeth are seen in the Moroccan Mosasaurus beaugei, just make the edges between the facets more protruding and facet surfaces more concave… It is important to demonstrate that these ridges are indeed additional serrated carinae, but not flutes, like those of some theropod dinosaurs (Paronychodon etc.)
It's not clear if we can answer this question. The ridges look like carinae and are serrated like carinae, and so their structure says could be homologous.
On the other hand, they occupy the position that facets/ridges/flutes do in other mosasaurs, so position argues that these are homologous to prisms/facets/ridges.
It is possible that the same genes that are responsible for carinae are also responsible for creating the prisms/ridges/flutes, in which case these structures are homologous with both.
It is unclear why the second tooth is obliquely photographed in apical view so that its cross-section and pattern of ridges and carinae distribution are incompletely seen.
The tooth is pointed with the base emerging vertically. The tip hooks back, so that the anterior of the tooth is visible in apical view.
Furthermore, in the second tooth, there are only two accessory carinae (vs four in the first) – this variation should be discussed.
It is pointed out in the manuscript.
In summary, I recommend:
adding close-up figures of serrations on both main and accessory carinae.
We have added a closeup figure of the first and second tooth.
adding remarks on the variation of the number of accessory carinae and their distribution around crown circumference
We note the variation in the description
optional - adding discussion on supernumerary carinae in the broader context of other vertebrates
Have added some comments on this discussing accessory cusps and accessory ridges in squamates, plesiosaurs, dinosaurs.
Specific comments
Abstract. Line 27: “diversity of mosasaurid teeth is much higher than that of plesiosaurs, ichthyosaurs, or marine mammals…” – I cannot agree regarding marine mammals - basal cetaceans have teeth of complex morphology with multiple accessory denticles and complex enamel ornamentation; furthermore the teeth of some pinnipeds, like crabeater seal, have a very complex morphology, clearly unparalleled by mosasaurs.
Our argument is that tooth diversity is higher, not that complexity. is higher. e.g. the diversity of Rotifera is higher than for Hominidae, although most people would argue Hominidae are far more complex. There are some pretty complex teeth in basal whales like basilosaurs but extant odontocetes invariably have simple teeth. Of course, they have managed to expand into new niches in terms of toothless forms like beaked whales and baleen whales, so it might not be a totally fair comparison, but the functional diversity of mosasaur teeth seems to be higher than for either toothed whales or pinnipeds.
Line 34: extra The in the first sentence
deleted
Line 61: Some reference is needed here to support the authors' statement e.g. https://doi.org/10.1073/pnas.110152510 or anything else appropriate to the context.
Added papers discussing relative abundance of whale species in Hawaii, Greenland, and the Mid-Atlantic Ridge.
Line 188: add reference where the “silky” enamel surface of prognathodontine mosasaurines is described.
Reference added
Lines 195-199: Correct figure captions, commas, tot dashes between the labels otherwise you have labial views on all pictures between a–f. and throughout. Also anterior, not “anteiro”.
Fixed
Figure 5 is uninformative, as it is pure speculation based on a very fragmentary specimen. I recommend not including it in the MS
Its simply meant to provide an approximate idea of the size of the animal based on the jaw fragment; its larger than animals like Carinodens but smaller than Mosasaurus. Given that size is one of the biggest predictors of diet (larger marine animals eat larger prey) the size is relevant to understanding its ecology.
Line 219. Format of the section heading.
Fixed
Line 255: correct to numerical citation. Also, the cited paper is not relevant to the context, as Zverkov & Pervushov (2020) described only a vertebra, and wrote nothing about teeth. Probably, the authors wanted to cite another paper, on pliosaurid teeth - https://doi.org/10.1111/pala.12367
Replaced this citation
Line 271-272: “such as” repeated. Rephrasing recommended
fixed
Line 281: “diversity of mosasaurid teeth stands in marked contrast to the more limited morphological diversity seen in their modern analogues, cetaceans and pinnipeds…” – this is not entirely true, as also commented on the abstract above - basal cetaceans have teeth of complex morphology with multiple accessory denticles and complex enamel ornamentation; furthermore the teeth of some pinnipeds, like crabeater seal, have a very complex morphology, clearly unparalleled by mosasaurs.
So authors can say that mosasaurids occupied more feeding guilds than modern marine mammals, but they should also acknowledge that there are tooth morphotypes and feeding adaptations among modern mammals not paralleled by mosasaurs.
That is the point we are trying to make: Globidens is occupying a niche that no whale occupies, although obviously a Basilosaurus molar is far more complex than any mosasaur tooth. You could conceivably argue that lack of teeth for filter feeding/suction feeding is something they didn’t do, however. We have added the line: “Although mammal teeth reach a level of complexity not seen in mosasaurs, mosasaur teeth arguably exhibit a wider range of morphologies and perform a wider range of functions than the teeth of marine mammals.”
Line 287: Title caption. I suppose ‘Conclusions’ should be here.
Fixed